# From CAD Models to Soft Point Cloud Labels: An Automatic Annotation Pipeline for Cheaply Supervised 3D Semantic Segmentation

Galadrielle Humblot-Renaux *, Simon Buus Jensen and Andreas Møgelmose

Visual Analysis and Perception Lab, Aalborg University, 9000 Aalborg, Denmark; sbje@create.aau.dk (S.B.J.); anmo@create.aau.dk (A.M.)
* Correspondence: gegeh@create.aau.dk

**Abstract:** We propose a fully automatic annotation scheme that takes a raw 3D point cloud with a set of fitted CAD models as input and outputs convincing point-wise labels that can be used as cheap training data for point cloud segmentation. Compared with manual annotations, we show that our automatic labels are accurate while drastically reducing the annotation time and eliminating the need for manual intervention or dataset-specific parameters. Our labeling pipeline outputs semantic classes and soft point-wise object scores, which can either be binarized into standard one-hot-encoded labels, thresholded into weak labels with ambiguous points left unlabeled, or used directly as soft labels during training. We evaluate the label quality and segmentation performance of PointNet++ on a dataset of real industrial point clouds and Scan2CAD, a public dataset of indoor scenes. Our results indicate that reducing supervision in areas that are more difficult to label automatically is beneficial compared with the conventional approach of naively assigning a hard "best guess" label to every point.

**Keywords:** 3D semantic segmentation; automatic labeling; soft labels; point clouds; deep learning

## 1. Introduction

Point cloud semantic segmentation is the task of classifying each point based on categories of interest (e.g., car, lamp, roof, person). It provides a dense, 3D description of the environment, with wide-ranging applications ranging from autonomous driving [1], augmented reality [2], and remote sensing [3] to the medical [4] or construction industries [5]. Recent advances in point cloud understanding have been largely driven by deep architectures that directly consume raw point clouds [6]. However, training deep point cloud segmentation models requires large quantities of labeled samples for supervision. Due to their sparse and irregular structure, 3-dimensional point clouds are much more laborious to manually annotate than images, making dataset creation impractical at a large scale. Current research is therefore limited to a small number of well-known general-purpose benchmarks [3]. In order to foster progress in deep learning-based point cloud understanding, enabling researchers and practitioners to efficiently annotate their own point cloud datasets across varied domains is key.

In this work, we investigate how the point cloud labeling process can be automated by leveraging CAD models at training-time. CAD models are particularly well suited for guiding semantic segmentation annotations, as they convey detailed information on the expected shape and geometry of designed objects and, being commonly categorized by name or shape, also implicitly carry the object description. The availability of CAD models at training-time is not unrealistic in practice, as large collections of 3D models are widely available (e.g., https://3dwarehouse.sketchup.com/ (accessed on 13 July 2023 ), https://grabcad.com/ (accessed on 13 July 2023)). For instance, thousands of 3D models of vehicles are available in the large-scale crowd-sourced repository Trimble 3D warehouse [7] and have successfully been used to label training images for autonomous driving [8,9]. In production applications, CAD models

are the industry standard for representing design parameters and requirements and are thus commonly provided by the manufacturer; many industry-grade CAD models of parts are also publicly available (e.g., https://www.traceparts.com/ (accessed on 13 July 2023)). For general scene understanding, the release of curated databases with high-quality 3D models of common objects [10] has motivated the creation of datasets such as ObjectNet3D [11], PIX3D [12], and Scan2CAD [13], which provide correspondences between images or point clouds and 3D shapes matched to the objects of interest. We consider two scenarios in our work: (1) indoor scene understanding, where we use the public dataset Scan2CAD [13] to label ScanNet point clouds [14], and (2) industrial workpiece segmentation, with CADs and point clouds collected from a real facility.

In our pipeline, we assume that at training-time, 3D models have already been selected from a database and fitted to the point clouds - the 3D model retrieval and alignment steps are beyond the scope of this work (see, e.g., [15,16] for work in this direction).

Our work tackles the subsequent problem: *given a dataset of point clouds with corresponding CAD models for objects of interest, how should each point be labeled in order to supervise the training of a deep segmentation network?* Assigning a correct label to each point based on its position in relation to a 3D model is not a trivial task: in practice, the scan-to-model correspondences are often only approximate, either due to retrieval errors, poor model fits, or scanning imperfections (noise, outliers, and other point cloud artifacts). To account for this ambiguity, we propose a *soft* label assignment method, which propagates model fitting errors to the estimated labels as soft object scores (visualized in the far-right of Figure 1). This score can either be used as a basis for soft semantic labeling or thresholded to obtain weak (middle-right) or standard one-hot labels (middle-left). We evaluate these three labeling schemes on two challenging datasets and show that incorporating ambiguity via soft labels when training a PointNet++ model improves segmentation performance while allowing the model to learn a useful notion of point-to-CAD fitness, which can be used at inference time to discard points deviating from the object.

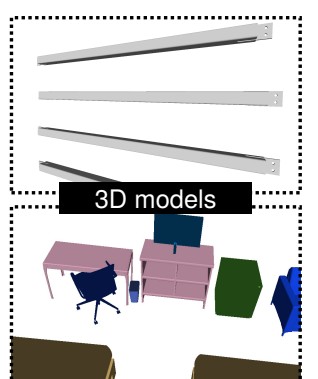
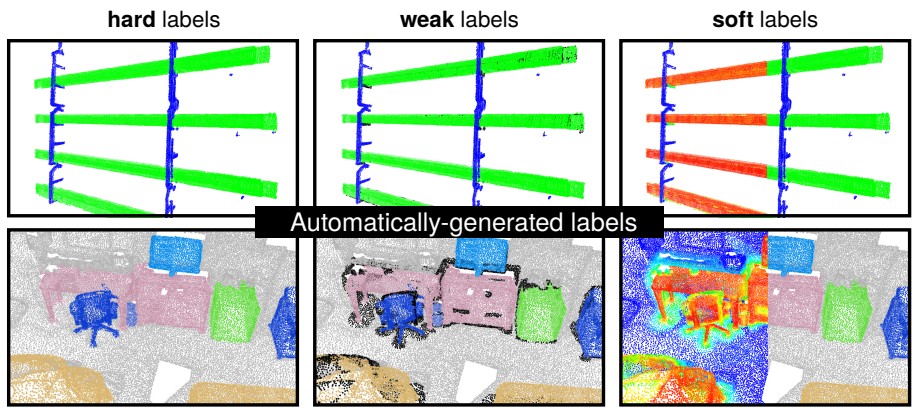

**Figure 1.** Given 3D models (**left**) fitted to a raw point cloud, our method generates point-wise semantic labels which can be used as cheap training data (**right**) for deep point cloud segmentation, either as soft, weak or dense hard labels—see Section 3 for details. We consider two use cases spanning different domains: steel workpiece extraction and indoor scene understanding.

To summarize, the main contribution of this paper is a semantic point cloud annotation pipeline that automatically assigns a semantic class and object score to each point based on CAD models while taking into account *label ambiguity* caused by model fitting errors and point cloud imperfections. Furthermore, unlike existing approaches for efficient point cloud labeling [17–20], our approach is not tailored to a specific dataset, domain, geometry, or set of classes. This pipeline has the potential to greatly speed up the process of generating new point cloud segmentation datasets across a wide range of applications where (approximate) CAD models of objects of interest are available at labeling-time but too expensive to retrieve and fit at test-time.

## 2. Related Work

There are broadly two areas relevant to this paper: (1) leveraging 3D models to automate point cloud labeling (Section 2.1, Section 2.2 and Section 2.3 below) and (2) learning segmentation from soft labels (Section 2.4). Our work lies at the intersection of these, which has not been explored in prior work.

### 2.1. Model-Based Segmentation

The segmentation of point clouds based on geometric primitives is well studied [21]. The review in [22] outlines the main methods and applications in this direction. For instance, iterative model fitting procedures based on energy minimization [23] or hypothesis testing [24,25] have been developed in order to segment planar and cylindrical surfaces in 3D scans of buildings.

However, these classical approaches do not consider object semantics, are often restricted to a small set of simple shapes, and require the expected shape to be known at runtime. In practice, despite the wide availability of 3D models, it is common for the model of an object to be unavailable or to no longer correspond to the real geometry of a manufactured object. Our approach is therefore model-based only at training time, when semantic segmentation labels are being generated. At inference time, the segmentation is model-free, requiring only a raw point cloud as input to the segmentation network. Furthermore, our method treats 3D CAD models as meshes, not parametric models, and thus can handle arbitrarily complex geometries. Note that we use the terms 3D model/CAD model interchangeably throughout the paper.

### 2.2. Cheap Point Cloud Labeling

To obtain dense semantic labels while reducing the burden on the annotator, many point cloud labeling tools and automation schemes have been proposed [26]. A promising direction is to leverage cross-modal information. For instance, in the context of autonomous driving, [19,20] show how semantic or object detection labels from images can be transferred to LiDAR scans. However, this assumes that an RGB modality is available, only considers a single viewpoint, and is restricted to the fixed set of classes learned by pre-trained image models. Other works require sparse user input (e.g., small number of labeled points in each scan [20,27], 3D bounding boxes [28]), and refine or propagate these coarse labels into dense point-wise annotations. Compared with these works, our approach is fully automatic; it does not rely on existing annotations or on a human in the loop.

Rather than labeling every point, an alternative is to learn from incompletely labeled point clouds under a weak supervision scheme. Remarkable performance has been obtained with only a tiny fraction of labeled points [29] or scribbles [30]) by enforcing neighborhood constraints during learning. While we take a naive, fully supervised approach in our evaluation, the weak labels generated by our annotation scheme (middle-right of Figure 1) are compatible with incomplete supervision training schemes such as [29].

### 2.3. CAD Models and 3D Shape Priors for Dataset Generation

The potential of 3D models as a modality for cheap training data generation has been explored in the image domain in previous work [8,9,31]. For instance, the method in [9] uses the contours of 3D car models to produce pixel-wise semantic annotations of autonomous driving images. Ref. [31] projects 3D models of built structures in a power plant onto panoramic images to generate panoptic segmentation labels. For remote sensing applications, ref. [32] uses 3D city models [33] to automatically generate a roof segmentation dataset from aerial images. These examples show the wide-ranging potential of existing 3D models for automating the labeling process. In contrast to these papers, our method operates on point clouds, which are often much more challenging and ambiguous to annotate than images due to their 3-dimensional, noisy, and unstructured nature.

In the point cloud domain, existing work leveraging CAD models for semantic segmentation labeling is scarce. 3D models have primarily been used as a means of generating

synthetic scans with "free" semantic labels [17,34–37]. In contrast, we tackle the problem of having to label *real* pre-existing point clouds, which were not sampled from a 3D model but from a real-world scene. In a similar spirit to our work, ref. [38] shows how existing Building Information Modelling (BIM) models can be used to automatically label point clouds of real buildings, train a deep segmentation model, and segment new point clouds at test-time for which a BIM model is not available. Compared with our labeling pipeline, the method in [38] relies on point-to-model distances with fixed dataset-specific thresholds and does not take into account label ambiguity.

### 2.4. Soft Labeling for Segmentation

When training a deep classification network, training labels are most typically encoded as "hard" one-hot vectors, with the target class having a probability of 1 and the rest 0. However, this treats classes as unrelated and labels as completely certain (the target class is considered absolutely correct, and the non-target classes are all completely and equally incorrect), which does not reflect the ambiguities often encountered during the labeling procedure [39]. For instance, when manually labeling a point cloud along the boundary between two objects, one may hesitate about whether a point should be assigned to one object or the other. Training a model on hard labels means that it will be harshly penalized for incorrect predictions, regardless of how ambiguous the input may be.

An alternative to one-hot encoding consists of "softening" labels such that the target class may not be completely certain (probability $\leq 1$) and the non-target classes may thus have a non-zero probability. Soft labeling is well-established in machine learning and computer vision as a form of regularization [40] and also as an effective way to express known relationships between classes (eg., similarity [41] or hierarchy [42–44]) or to capture natural ambiguity in the data (eg., at the borders of ground truth masks [45] or due to inconsistent/subjective labels from multiple annotators [46]). For image segmentation, soft labels have been shown to improve generalization, reduce mistake severity [44], and yield soft, informative predictions along object boundaries [45].

In the point cloud domain, the use of soft labels remains underexplored. Label smoothing has been shown to improve point cloud segmentation in recent work [47], but this treats all non-target classes as equally probable regardless of the data. Adjacent to soft labeling is pseudo-labeling, where model predictions are fed back as training data in a semi-supervised scheme [27]. In contrast, we generate soft point-wise labels from CAD models in a data-driven fashion.

Our motivation for taking a soft labeling approach for point cloud annotation is that while the category of *real physical objects* is unambiguous (we know exactly where a chair starts and ends), the semantic labels of points from *scans* of real objects can be ambiguous, even to a manual annotator; in a point cloud, it is not always clear where precisely a chair ends and the floor starts. Therefore, hard training labels may not always be the most appropriate way to supervise a point cloud segmentation network, especially if the labels are known to be inaccurate (which is our case due to the automated annotation procedure).

The novelty of this paper thus lies in the application of soft labeling to an approximate point cloud labeling setting, which is relevant across many concrete applications where deep learning-based point cloud segmentation is widely used but where manual label acquisition is prohibitively expensive, including remote sensing. To the best of our knowledge, this work is the first attempt at encoding the ambiguity of point-to-model correspondences as soft labels for training a deep segmentation network.

## 3. Automatic Labeling Method

Our method generates approximate point cloud labels, which can be used to cheaply train a deep point cloud segmentation network. Given an unlabeled point cloud and a set of fitted 3D models describing the objects of interest, our aim is to automatically assign a semantic label to every point. Points corresponding to one of the 3D models inherit its semantic class (e.g., "chair", "car"). The rest of the points are assigned to a "background" class. Based on

these fitted CAD models, our pipeline outputs a semantic category (e.g., "chair") per point coupled with an *object score* ranging from 0 to 1, where 1 indicates that the point most likely belongs to this object and 0 to the background. From this object score, we then propose to generate three types of automatic training labels, which we illustrate in Figure 2:

- **auto-hard** labels: every point is assigned a one-hot label (probability of 1 for the target class, 0 for the other classes)—this is the typical way that labels are defined for semantic segmentation. This annotation scheme does not take into account any uncertainty in the labeling process; during training, the model is equally penalized for wrong predictions in ambiguous regions (where labels are likely to be incorrect) as in easy-to-label regions.
- **auto-weak** labels: similar to the hard scheme, but points that are ambiguous are left unlabeled (visualized in black throughout the paper). During training, these points are not considered in the loss computation; the model is "free" to predict anything for these unlabeled points without being penalized for it.
- **auto-soft** labels: the target class and background probability are based on the object score. During training, the model is penalized more for misclassifying easy-to-label points than for misclassifying ambiguous points.

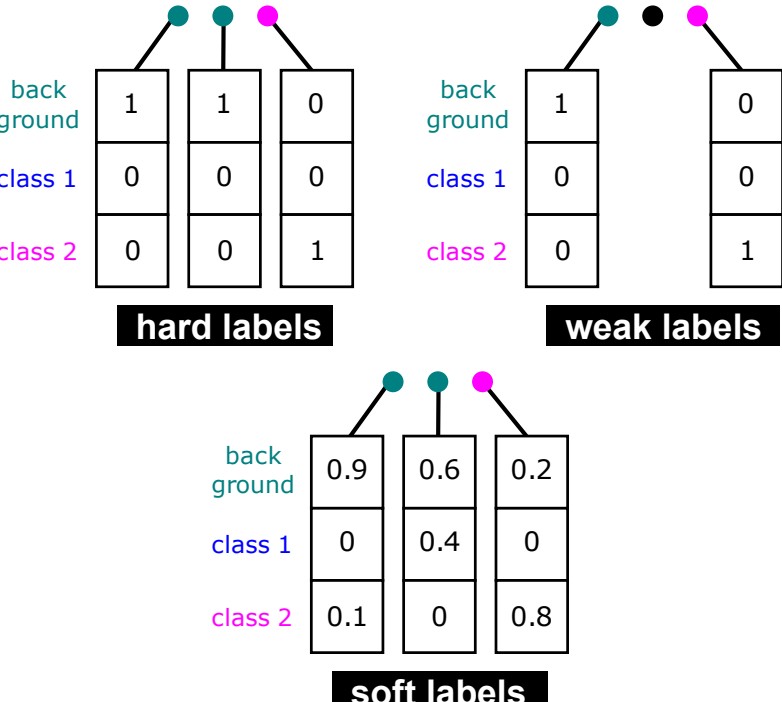

**Figure 2.** Illustration of the 3 types of training labels with a toy example featuring 3 points and 3 classes. Note that we use the same set of semantic classes across the 3 labeling schemes. In the hard and weak scheme, point labels are one-hot vectors, while for the soft scheme, point labels encode class membership probabilities.

The rest of this section explains how we go from CAD models to object scores to training labels. In Section 5, we systematically evaluate and compare these three labeling schemes.

### 3.1. Divide and Conquer

Figure 3 illustrates the first step in our pipeline, which tackles the question: **out of the 3D models in the scene, to which object does each point most likely belong?** For this, we split the point cloud into *sections* (one per model) based on point-to-model distances, as described in Algorithm 1.

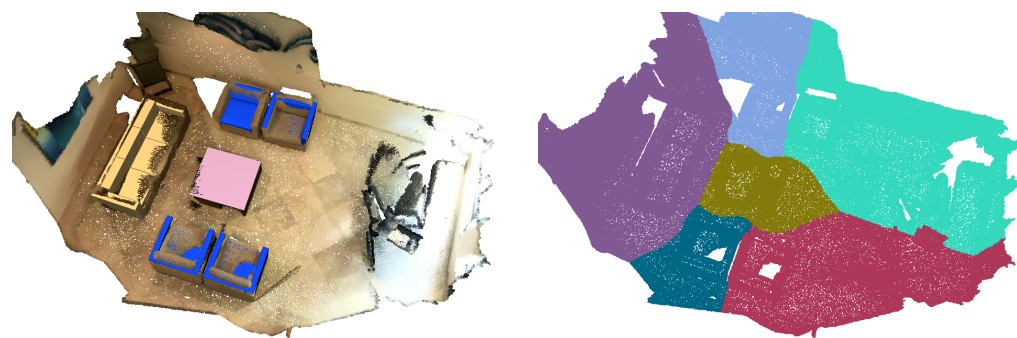

**Figure 3.** Sample from Scan2CAD [13] used to illustrate our method (cf. Section 4 for dataset details). (**Left**) Input point cloud and fitted 3D models, (**right**) point cloud sections which we label independently (arbitrary colors).

---

**Algorithm 1** Point cloud splitting algorithm

---

**Input:** Set of scan points $\mathcal{Q}$, sequence of $M$ 3D models $(O_1, ..., O_M)$
**Output:** A section assignment label $s \in \{1, ..., M\}$ for each point
  **for** each point $Q \in \mathcal{Q}$ **do**
    **for** $m \leftarrow 1$ to $M$ **do**
      $d_m \leftarrow$ shortest L2 distance between $Q$ and $O_m$       ▷ computed via raycasting
    **end for**
    $s \leftarrow \underset{x \in \{1,...,M\}}{\text{argmin}} \ d_x$
  **end for**

---

This point cloud splitting step allows us to reduce the problem to a binary label assignment per section, with the question now being: **which points belong to the model, and which points should be labeled as background?**

### 3.2. Zooming in on a Section

A naive approach could be to assign each point a label based on its distance to the CAD model. However, distance alone is often a poor indicator of whether a point belongs to the object vs. the background, especially in the case of noise or misalignment.

Taking Figure 4 as an example, points on the armrest (circled in pink) can be further away from the mesh than points on the floor lining the edge of the chair (in red)—yet the former should nevertheless be labeled as part of the object. Therefore, we propose to assign each point a holistic object score based on three individual scores, which are visualized in Figure 5 and explained below.

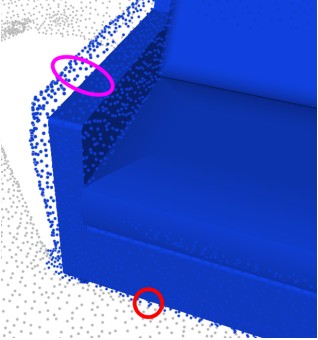

**Figure 4.** Close-up of a scan with manual semantic labels and fitted CAD model. Note that points belonging to the sofa (pink circle) are not necessarily closer to the CAD model than floor points (red).

We define $P$ as the set of $n$ points within the section, $M$ as the mesh surface of the CAD model, and $M_{closest}$ as the set of $n$ points on $M$ closest to $P$. Each score is a continuous value per point $\mathbf{p} \in P$ ranging from 0 (background) to 1 (object), as illustrated in Figure 5d.

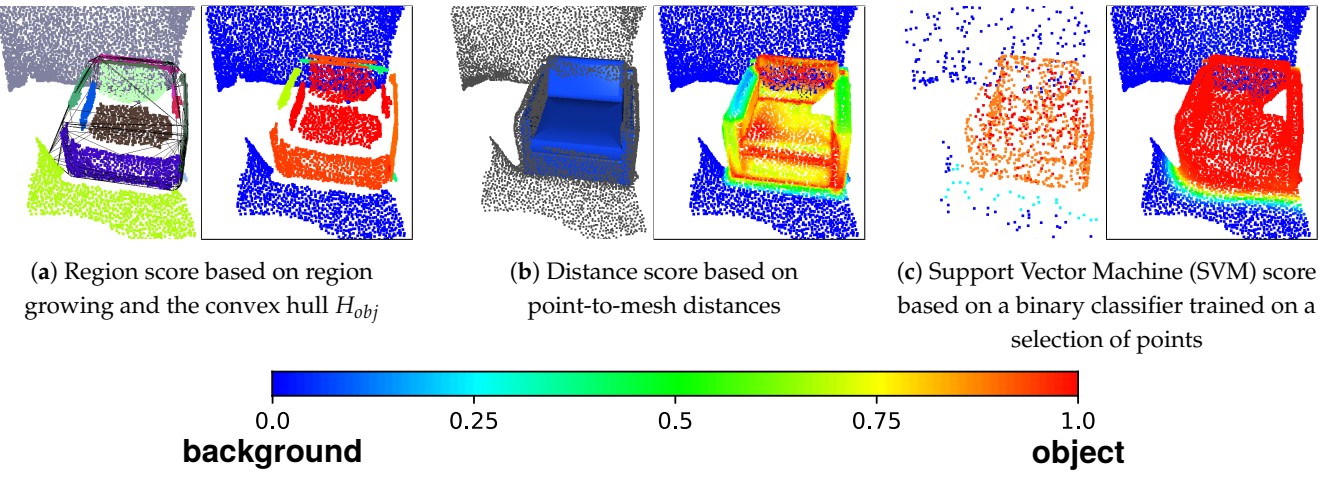

(**a**) Region score based on region growing and the convex hull $H_{obj}$

(**b**) Distance score based on point-to-mesh distances

(**c**) Support Vector Machine (SVM) score based on a binary classifier trained on a selection of points

0.0      0.25      0.5      0.75      1.0

**background**                       **object**

(**d**) Color map for object scores.

**Figure 5.** Visualization of the 3 scores (framed in black) for an armchair section.

### 3.2.1. Region Score ($R_{score}$)

We use the region growing segmentation algorithm [48] to find groups of points likely to belong to the same surface. Each group is then scored based on the proportion of points lying inside the object's convex hull $H_{obj}$. We take $H_{obj}$ as the convex hull enclosing the set of points in $P$ closest to $M_{closest}$. A group's score thus ranges from 0 (no points inside $H_{obj}$) to 1 (all points). This region-level score assignment can allow individual points that are not necessarily very close to the model (e.g., due to fitting errors) to be assigned a high score if they belong to a region that largely intersects with $H_{obj}$. Both the convex hull and regions are shown on the left of Figure 5a. Note that points not satisfying the region's growing constraints are not assigned to any group, hence the gaps between regions. We leave the region score for these points undefined.

A well-established drawback of the region-growing algorithm is its sensitivity to constraint parameters. However, our method is flexible in terms of the number of clusters found, as long as object/background separation is achieved. If this is not the case with the initial parameters, the smoothing and curvature constraints are automatically loosened/tightened iteratively until at least two clusters are found.

### 3.2.2. Distance Score ($D_{score}$)

This is taken as the shortest unsigned distance $D(\mathbf{p})$ of a point to the mesh surface (shown on the left of Figure 5b). The distance is then mapped from a metric scale to a score based on a distance threshold $t$:

$$D_{score}(\mathbf{p}) = \begin{cases} 0, & \text{if } D(\mathbf{p}) > t \\ 1 - D(\mathbf{p})/t & \text{otherwise} \end{cases}$$

The choice of an appropriate value for $t$ is challenging, as it varies per dataset and per section - setting it too large leads to over-segmentation, while setting it too low results in a low score for any point slightly deviating from the CAD model. We bypass the need for manual threshold selection by setting $t$ adaptively based on the region score as follows: $t = \max\{D(\mathbf{p}) \mid R_{score}(\mathbf{p}) > 0.5\}$—that is, we take $t$ as the maximum distance among points in regions that are more likely to belong to the object than the background.

### 3.2.3. SVM Score

The location of a point alone can be a strong indicator of its class—for instance, in Figure 5, all points above the ground and in front of the wall belong to the armchair. We therefore introduce a statistical model-based score which assigns a probability to each point by estimating a non-linear decision boundary separating the object and background. We train a C-support vector machine (SVM) [49,50] with a radial basis function kernel on a selection of points that can confidently be assumed to belong to the object or background. The sets of points used for training are outlined in Table 1. As object points, we consider $M_{sampled}$ a random set of 1000 points uniformly sampled directly from the mesh surface and $P_{closest}$, the set of points in P closest to $M_{closest}$. As background points, we consider points with a low region score as well as those lying outside of $H_{mesh}$, the convex hull of the mesh scaled by a factor of 1.5 (leaving a generous margin for CAD alignment errors). Training points are assigned a per-sample weight $w$ which scales the regularization parameter C: subsets which we are most confident about ($M_{sampled}$, and points outside of $H_{mesh}$) are given a higher weight, while the other two subsets are considered more ambiguous.

**Table 1.** Sets of points used to fit the binary SVM classifier.

| Class | Points | $w$ |
|:---:|:---:|:---:|
| **object** | $M_{sampled}$ | 10 |
| | $P_{closest}$ | 5 |
| **background** | $\{\mathbf{p} \mid R_{score}(\mathbf{p}) < 0.25\}$ | 1 |
| | $\{\mathbf{p} \mid \mathbf{p} \text{ outside of } H_{mesh}\}$ | 10 |

Since the fitting time of kernel-based classification scales poorly to an increasing number of points, we randomly sample a fixed set of points (1000 in our experiments) as training data for each of the two classes. An example set of training points is shown at the left of Figure 5c, color-coded based on $w$. Note that we directly use point 3D coordinates as features-the aim being to model the spatial area occupied by the object of interest. The fitted model is then applied to the full set of points in the section. We obtain a probabilistic score from the decision function via Platt scaling, which we visualize on the right of Figure 5c. Here, the SVM decision boundary separates the armchair from the wall and floor.

### 3.3. From Scores to Labels

We compute an overall object score $c \in [0, 1]$ for each point within a section by taking the average between the region, distance, and SVM scores. For points where the region score is undefined, only the distance and SVM scores are averaged. We conduct an ablation study in Section 5.2, isolating the benefit of each score. Figure 6 visualizes the result after combining the object score across sections.

In the **hard** labeling scheme (middle of Figure 6), we label the point as background if it has an object score $c < 0.5$; otherwise, it inherits the category of the 3D model. In the **weak** labeling scheme (left of Figure 6), only points that are highly likely to belong to either class are labeled; ambiguous points ($0.25 < c < 0.75$) are left unlabeled (visualized in black). In the **soft** labeling scheme (right of Figure 6), we directly use $c$ as the class probability for the object's category ($1 - c$ for the background and 0 for the other classes in the point cloud).

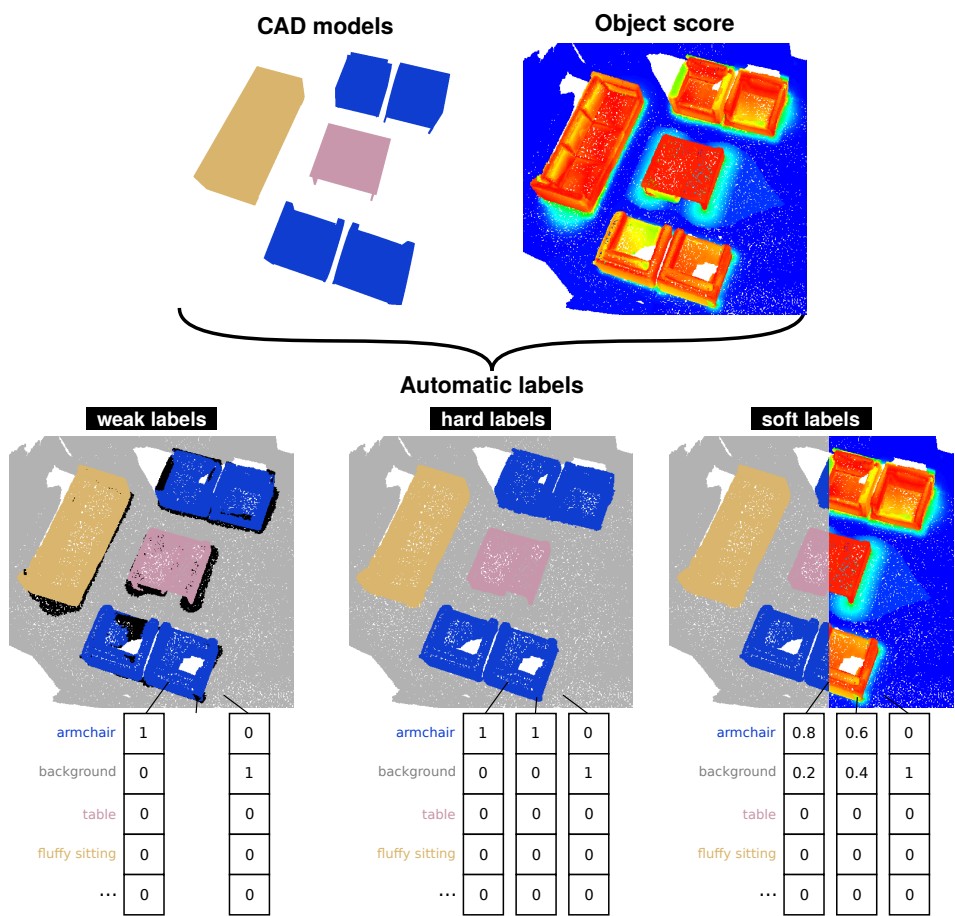

**Figure 6.** Automatic labelling output (cropped for readability). We show the class probability vectors for 3 points around an armchair to illustrate the differences between the weak, hard and soft labelling schemes. Note that the target class is based on the category of the closest CAD model, while the probability of the target class vs. background is based on the object score. Black points in the weak scheme are unlabelled. See Figure 5d for the object score colormap.

## 4. Data

We develop and evaluate our method on two datasets: Beams&Hooks, a proprietary dataset of steel beam laser scans from a real industrial use case, and Scan2CAD [13], a public dataset combining the RGB-D indoor room scans of ScanNet [14] and the 3D shape database ShapeNet [10]. Table 2 gives an overview of the dataset's statistics.

**Table 2.** Overview of the 2 datasets. The symbol # stands for "number of".

| *Dataset* | #Classes | #Scans | Avg. #Points per Scan | Unique CADs | Avg. #CADs per Scan |
|---|---|---|---|---|---|
| Beams&Hooks | 2 | 775 | 798,315 | 149 | 5.2 |
| Scan2CAD | 13 | 1506 | 147,941 | 3049 | 9.4 |

### 4.1. Beams&Hooks

Beams&Hooks is an industrial point cloud dataset of steel workpieces suspended by hooks, collected over a 3-month period in a robotic painting facility (see [51,52] for existing work on the topic). The steel workpieces are scanned as they enter the painting area, in order to identify the surfaces to paint and generate the robot's control sequence. As the hooks supporting the workpieces should not be painted by the robot, they need to be identified via 2-class point cloud segmentation (each point being either *workpiece* or *hook*).

The point cloud acquisition setup consists of six custom-built laser line scanners placed around an overhead conveyor carrying the steel workpiece, such that all viewpoints are covered. Each laser line scanner consists of a laser line projector and a Basler monochrome camera with a resolution of $1280 \times 1024$ operating at 75 FPS. Line scans are captured continuously as the workpiece is conveyed through the scanning area. A 3D profile of the workpiece's surface is generated by combining line distance measurements with the known geometry and positions of the projector and camera. This scanning set-up has an accuracy of $\pm10$ mm (with workpieces measuring on average 5.5 m long, 0.2 m high, and 0.1 m deep).

Our Beams&Hooks dataset only includes scans for which the as-planned 3D CAD model of each piece is provided by the manufacturer and fitted to the 3D scan—some examples are shown in Figure 7. The workpieces vary in their geometry (straight or tapered), in their cross-section (primarily I-beam or tubular), in their supports (type, number, and location), as well as their dimensions.

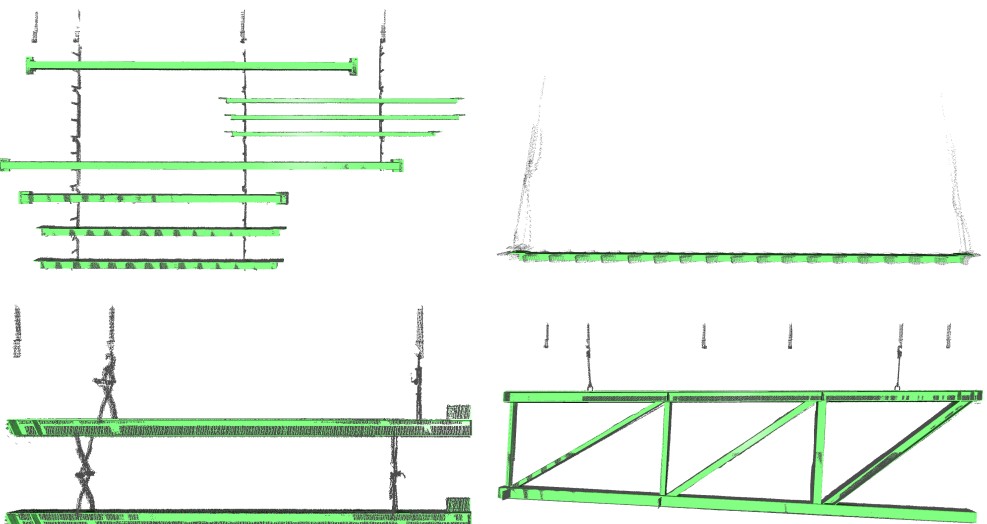

**Figure 7.** Four samples from the Beams&Hooks dataset, showing the raw point cloud (gray) and fitted CAD models (green). Note that the hooks may carry many workpieces at once (**left**), or a single workpiece (**right**).

Note that while the retrieved CAD model matches the scan in most cases, this data presents an interesting set of challenges for point cloud understanding: it suffers from outliers and incompleteness caused by sensor noise and metal reflectivity, misalignment and stitching errors, wave-like patterns caused by the movement of the piece moving during the scanning process, and highly variable point densities.

To obtain "true" ground truth segmentation labels for evaluation, we asked an expert human annotator to provide high-quality manual annotations for a subset of 50 samples. This test set is curated to cover the diversity in workpiece types, hook attachments, and layouts across the dataset.

### 4.2. Scan2CAD

This dataset acts as a link between ScanNet's RGB-D scans [14] (which also come with manual semantic segmentation labels) and the 3D model database ShapeNet [10]. ScanNet's 3D scans were collected using a Structure depth sensor (resolution $640 \times 480$)—we refer to the original ScanNet paper [14] for details on the acquisition set-up. ShapeNet is a well-known curated database of 3D shapes featuring a large variety of vehicles, furniture, and household objects organized by category; for instance, it contains over 850 models of bathtubs. For over 1500 ScanNet scans, Scan2CAD provides a list of corresponding ShapeNet models along with

their 9 DoF pose in the scene. We refer to the original paper [13] for a description of the alignment procedure.

Compared with Beams&Hooks, the point clouds in this dataset are much "cleaner" (uniformly sampled, less noise). The challenge for automatic labeling rather lies in the large semantic diversity, object clutter, and complex scene geometries, coupled with much looser 3D model-to-point cloud correspondences. Indeed, upon further inspection, we found many instances of missing, misaligned, or inaccurate CAD models (e.g., a chair model fitted to a toilet, a bed fitted to a sofa).

To obtain "true" ground truth labels for evaluation, we take advantage of the crowd-sourced semantic labels in ScanNet [14]. However, we cannot use them out-of-the-box, since ScanNet labels are incompatible with ShapeNet categories, and not every annotated piece of furniture in the point clouds has a corresponding Scan2CAD-fitted model. For comparison with our automatic labeling scheme (which assigns labels based on the CAD category), we therefore generate ground truth semantic labels for Scan2CAD by mapping ScanNet annotations and ShapeNet CAD model categories to a common class definition—as shown in Figure 8.

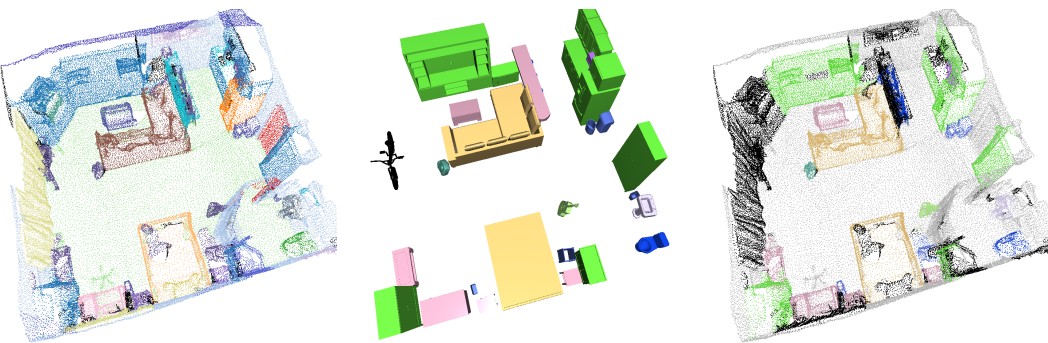

**Figure 8.** Example showing how we generate a "true" ground truth sample (**right**) for the Scan2CAD dataset using ScanNet labels (**left**) and fitted ShapeNet models (**middle**). CAD models are shown color-coded according to our class definition.

We define 13 semantic classes for our task: a "background" class for flat surfaces in the room (e.g., ceiling, floor, door) and 12 types of objects: appliances, fluffy sitting (sofa/bed), basket, instrument, lamp, bookshelf, chair, table, electronic device, water, bag, cabinet. Object semantic classes from ScanNet that have no corresponding class in ShapeNet (e.g., counter, box, person) or have no Scan2CAD-fitted model are left unlabeled (black in Figure 8). Since our labeling method assumes that a CAD model is available for each object (and would wrongly label object points as "background" otherwise), we discard these points in our experiments.

## 5. Experiments

We first present the overall evaluation procedure in Section 5.1. In Section 5.2, we evaluate the quality-vs-effort trade-off of labels generated by our automatic annotation scheme. We also zoom in on the proposed labeling scores with an ablation study. In Section 5.3, we train a point cloud segmentation network on these approximate labels and compare performance across the three schemes, followed by some more detailed insights into the benefits of a soft label approach.

### 5.1. Evaluation Procedure

The evaluation is divided into two experiments. In Experiment 1, we directly compare the automatic point cloud labels with manual labels. In Experiment 2, we evaluate the predictions of PointNet++ trained on automatic labels by comparing the predictions with manual labels. For both experiments, we used manual labels as ground truth and reported

the following segmentation metrics. All metrics are computed per point cloud and averaged across the evaluation set.

- **overall accuracy (OA)**—the proportion of correctly classified points (note that this metric can obfuscate poor performance in minority classes)
- **mean accuracy (mACC)**—class-wise accuracy, averaged across all classes
- **mean intersection over union (mIoU)**—class-wise IoU, averaged across all classes
- **macro-F1**—class-wise F1 score, averaged across all classes

In addition to overall performance, we investigate segmentation performance along object boundaries, since these are the most ambiguous to annotate. We adopt the boundary-specific evaluation metrics from [53] mIou@boundary and mIou@inner, where a point is considered a boundary if it has one or more points with a different ground truth label in its neighborhood (radius of 0.1 m in our experiments). Figure 9 gives an example of boundary points on our two datasets.

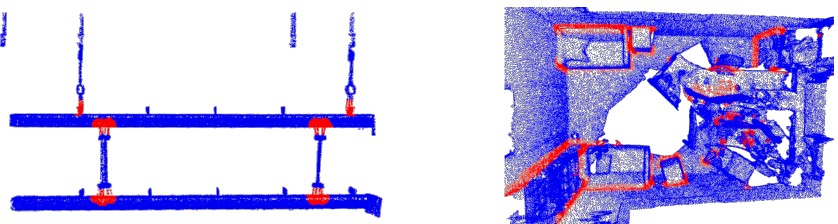

**Figure 9.** Visualization of boundary (red) vs. inner (blue) points used to compute boundary-specific metrics on a point cloud sample from Beams&Hooks (**left**) and ScanNet (**right**).

### 5.2. Experiment 1: Automatic vs. Manual Labels

Aligning with prior work [20], we compare the approximate labels generated by our proposed automatic labeling method with "true" ground truth labels obtained via manual annotation.

#### 5.2.1. Qualitative Results

Figure 10 shows a side-by-side comparison of labeled samples. The second two rows are clear examples of CAD fittings only being approximate (e.g., cushions/pillows protruding from the couch and bed models), yet our method outputs convincing hard labels while identifying these areas as being ambiguous in the soft and weak schemes.

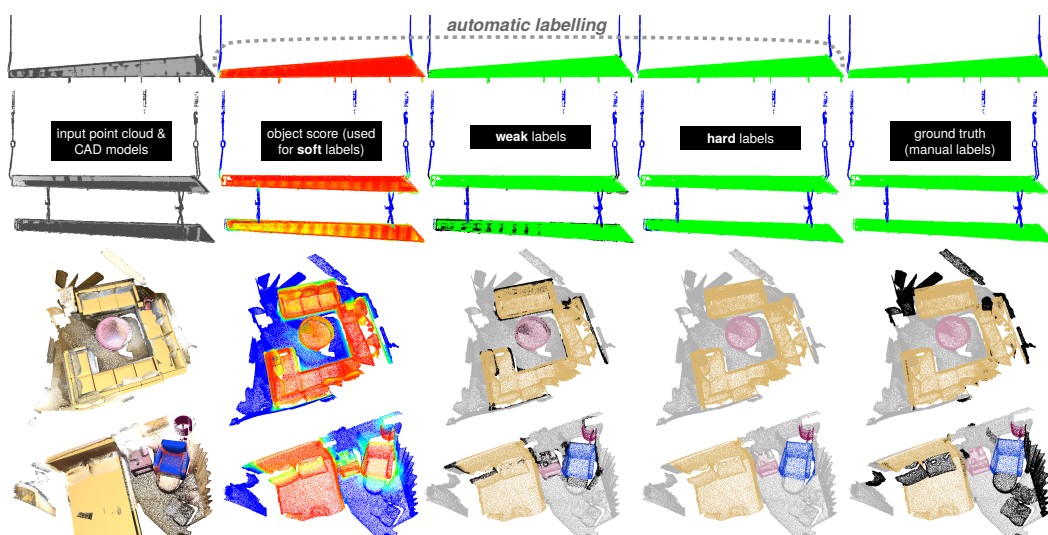

**Figure 10.** Qualitative comparison of the approximate labels and "true" ground truth. See Figure 5d for the object score colormap.

### 5.2.2. Quantitative Results

We evaluate the label quality in Table 3 and Figure 11. Note that results for the soft labeling scheme are not shown in Table 3, since the soft labels are identical to hard labels after applying argmax.

**Table 3.** Comparison of our approximate labels with manual ground truth labels. Scores are in %.

| | | | | | | mIoU | |
| labels | % labeled | OA | mACC | macro-F1 | overall | @bound. | @inner |
|---|---|---|---|---|---|---|---|
| **(a) Beams&Hooks (test set, 50 samples)** | | | | | | | |
| auto-hard | 100 | 99.07 | 97.81 | 96.21 | 93.30 | 86.80 | 93.16 |
| auto-weak | 95.02 | 99.52 | 98.97 | 98.37 | 96.95 | 91.88 | 96.81 |
| **(b) Scan2CAD (validation set, 312 samples)** | | | | | | | |
| auto-hard | 100 | 93.15 | 88.92 | 88.28 | 80.81 | 56.42 | 86.13 |
| auto-weak | 89.42 | 96.00 | 91.75 | 91.91 | 86.95 | 64.43 | 90.81 |

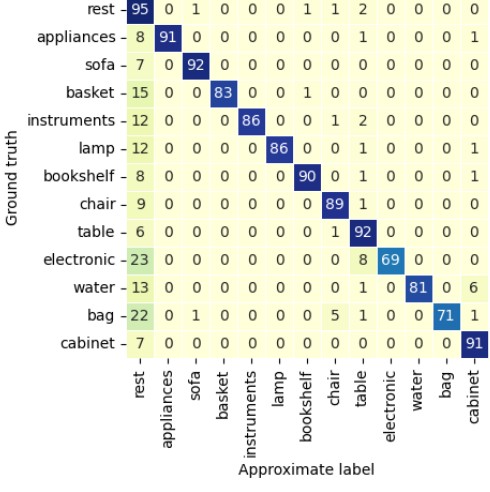

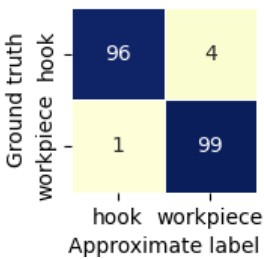

(**a**) Scan2CAD validation set  (**b**) Beams&Hooks test set

**Figure 11.** Confusion matrix of auto-hard labels vs. manual ground truth. Normalization is by row with class-wise recall in the diagonal; scores are in % ranging from 0 (light yellow) to 100 (dark blue).

Over 99% of Beams&Hooks points are correctly labeled with both the weak and the hard schemes. The gap between the two schemes widens for the Scan2CAD dataset, where a larger proportion of points are left unlabeled with the weak scheme. The boundary-specific metrics confirm the challenge of correctly labeling areas where objects intersect, especially for Scan2CAD, where there is a 30% gap in mIoU between boundary and inner areas.

The confusion matrix in Figure 11 examines label quality on a class-wise basis. For instance, 90% of "bookshelf" points in Scan2CAD are correctly labeled by our automatic method, and 8% of "electronic device" points are mislabeled as "table". Overall, our labeling method has the tendency to under-segment objects in Scan2CAD (highest recall for the "rest" class), while over-segmenting the workpiece in Beams&Hooks (lower recall for the hook). Small or thin objects (instrument, electronic, bag, basket) have the lowest class-wise recall in Scan2CAD, which is to be expected since for small objects, even small CAD fitting errors can lead to mislabeling of a large portion of points.

Lastly, in Figure 12 we examine labeling accuracy in relation to label object score. Points with scores around 0.5 (uncertain label) are the least likely to be correctly labeled in the hard scheme, while those approaching 0 (background) or 1 (object) are almost always labeled correctly. This shows two things: (1) the object scores produced by our automatic

labeling scheme (which we inject into soft labels as the object probability) are a reliable indicator of label "correctness" and (2) points left unlabeled in the weak scheme (score between 0.25 and 0.75) are indeed significantly less likely to be correctly labeled in the hard scheme than the rest.

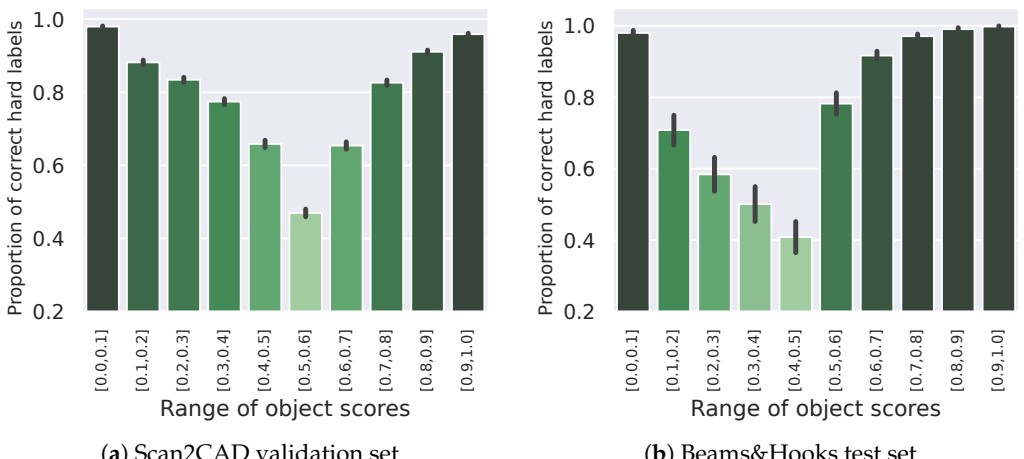

(**a**) Scan2CAD validation set                          (**b**) Beams&Hooks test set

**Figure 12.** We select subsets of points by label object score and report how many are correctly labeled on average by the hard labeling scheme. Error bars show standard error across samples.

### 5.2.3. Annotation Time

To quantify labeling effort, we asked the annotator of Beams&Hooks to record the time spent labeling each sample—the distribution of timings is shown on the left of Figure 13. Each point cloud took 42.8 min on average to manually annotate, with several taking over two hours—clearly demonstrating the prohibitive time and labor cost of manual point cloud annotation. We found that the manual labeling time did not so much depend on the number of CAD models or points as on the complexity and ambiguity of boundary areas where the hook attaches to the workpiece. In contrast, our automatic labeling scheme takes 31.8 s on average per input point cloud (on CPU@3.5GHz), scaling linearly both to the number of points and the number of CAD models. Note that we consider these timings to be an upper bound measure with ample room for improvement since our implementation was developed in Python with convenience and legibility in mind rather than computational efficiency.

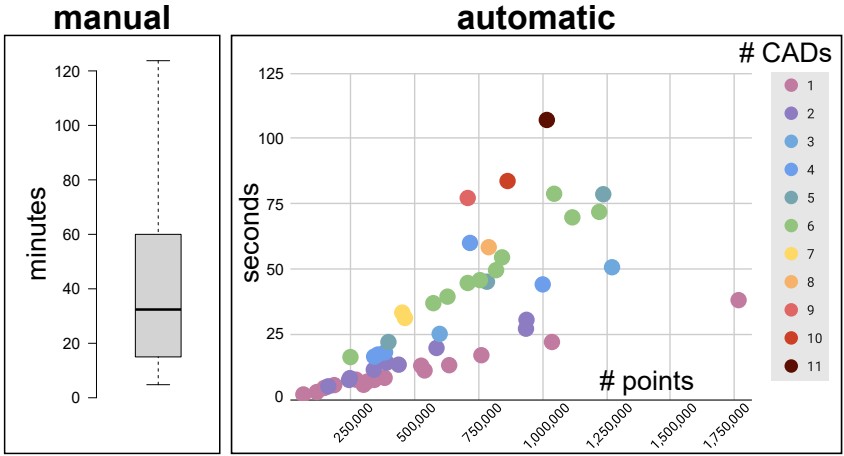

**Figure 13.** Time taken to annotate each sample in the Beams&Hooks test set. For the automatic labeling scheme, points are colored based on the number of fitted CAD models.

### 5.2.4. Ablation Study of Labeling Scores

We study the influence of individual scores (region, distance, and SVM score described in Section 3.2) on the label quality, following the same procedure as in the previous experi-

ment, where approximate labels are compared with manual ground truth labels. We report the results in Figure 14.

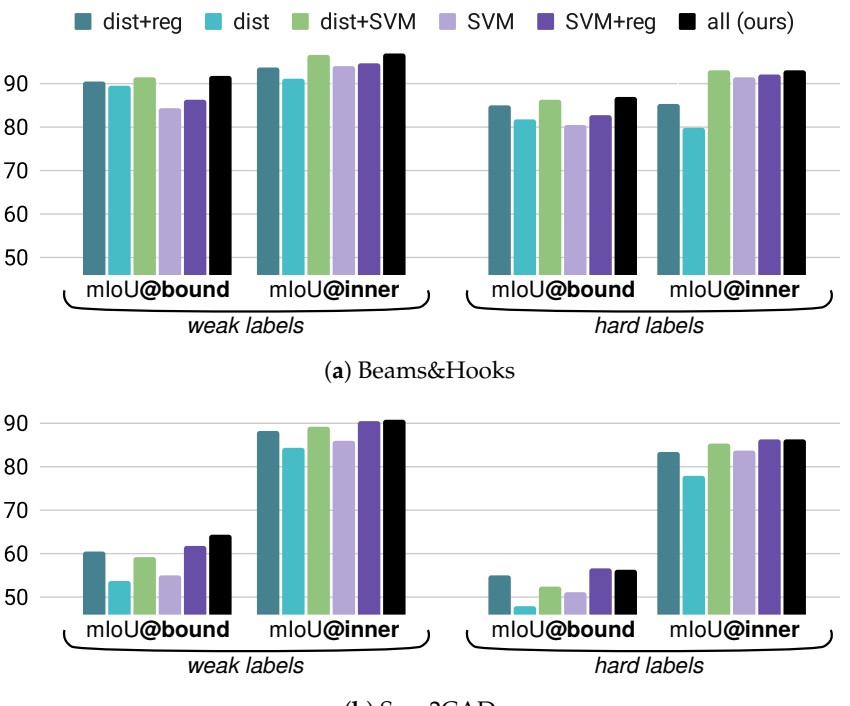

(**a**) Beams&Hooks

(**b**) Scan2CAD

**Figure 14.** Segmentation quality of weak and hard labels for different combinations of scores.

Distance as a standalone score yields the lowest mIoU@inner across both datasets, especially in the hard labeling scheme, as it causes object points that deviate from the 3D model (noise, outliers, misalignment) to be mistaken for the background class. The addition of the region score (dist + reg, SVM + reg) mostly brings an improvement for boundary areas in Scan2CAD, as it encourages crisp separation between groups of points belonging to the same surface. Conversely, the SVM score primarily improves segmentation in inner areas (especially for Beams&Hooks), as it learns a smooth, regularized decision boundary around the object and outputs high probabilities for points within it. Combining all three scores yields the best cross-dataset performance.

### 5.3. Experiment 2: Training a Segmentation Model

Here we evaluate the performance of a deep segmentation model trained on the 3 different types of approximate labels.

#### 5.3.1. Architecture and Hyper-Parameters

We apply the point-based network PointNet++ for deep point cloud segmentation [54]. It extends the pioneering model PointNet [55] with hierarchical feature learning and density adaptive layers, which are key for robustness to non-uniform sampling density (as present in the Beams&Hooks dataset). While PointNet++ has been out-performed by more recent architectures [6], it remains a common choice for benchmarking. We train PointNet++ with a KL divergence objective using Adam optimization [56] (initial learning rate of $10^{-3}$—aligning with [54]). Samples are fed to the network in shuffled minibatches of size 4. The best model is selected based on minimal validation loss.

#### 5.3.2. Dataset Splits

For Beams&Hooks, we set the manually annotated test set aside and split the rest of the dataset (725 samples) randomly into a training and validation set following an 80/20% split. For Scan2CAD, we use the official ScanNet training/validation split (1201/312 samples,

respectively). Since a test set is not publicly available, we use the validation set samples for evaluation (aligning with [54]), but with the true ground truth labels described in Section 4.2. Note that in both cases, the training and validation sets only contain automatic annotations: the model never "sees" a true label during learning, nor is any true label used for model selection.

### 5.3.3. Data Pre-Processing

We normalize input point clouds via zero-centering. Since PointNet++ only takes a fixed small set of points as input, during training we randomly sample a subset of 8000 points on the fly from the whole point cloud. At inference time, we repeatedly sample 8000-point subsets until the whole point cloud has been processed.

### 5.3.4. Results

We evaluate model performance in Table 4. Since true (manual) ground truth labels are available for the whole Scan2CAD dataset (cf. Section 4.2), we train a "manual baseline" model as an upper baseline (italicized in Table 4). The manual baseline model's mediocre performance (mIoU of 41.56%) underlines the difficulty of this segmentation task, especially for long-tailed/underrepresented classes. We therefore also report the model's ability to discriminate between the background and objects (regardless of class) by mapping 13-class predictions to 2 classes; results for the 2-class performance are given in Figure 15.

Interestingly, across both datasets, we find that training PointNet++ on auto-hard labels yields consistently worse performance than auto-weak or auto-soft labels, with a performance gap between the auto-hard vs. auto-weak/soft schemes that is significantly wider than between the auto-weak vs. auto-soft schemes. For Scan2CAD, the soft and weak models even outperform the manual baseline model (trained on manual semantic labels) across all class-balanced metrics (mACC, macro-F1, and mIoU). The weak and soft models achieve similar accuracy but mostly differ in IoU scores. Most notably, training on soft vs. hard labels improves boundary mIoU by almost 3% on Beams&Hooks, and inner mIoU by 2% on Scan2CAD. Comparing the 2-class confusion matrices in Figure 15, we find that for both datasets, the soft PointNet++ model achieves the most balanced performance between classes and most closely approaches the label performance from Experiment 1 (Figure 11).

**Table 4.** Performance of PointNet++ for different types of training labels, with the **best** and worst scores highlighted. All the models are evaluated against manual labels as ground truth.

| (a) Beams&Hooks (test set, 50 samples) | | | | | | |
|---|---|---|---|---|---|---|
| *training labels* | **OA** | **mACC** | **macro-F1** | | **mIoU** | |
| | | | | overall | @bound. | @inner |
| auto-hard | 97.37 | 95.51 | 94.78 | 91.36 | 75.74 | 93.66 |
| auto-weak | **99.27** | 96.35 | **96.97** | **94.26** | 76.64 | **97.86** |
| auto-soft | 99.13 | **96.67** | 96.58 | 93.68 | **78.64** | 96.63 |

| (b) Scan2CAD (validation set, 312 samples)—13-class | | | | | | |
|---|---|---|---|---|---|---|
| *training labels* | **OA** | **mACC** | **macro-F1** | | **mIoU** | |
| | | | | overall | @bound. | @inner |
| *manual* | ***87.40*** | *47.63* | *47.97* | *41.56* | *25.57* | *45.44* |
| auto-hard | 86.06 | 46.85 | 46.59 | 40.44 | 25.99 | 44.12 |
| auto-weak | 86.78 | **48.97** | **49.53** | 42.07 | **26.08** | 45.67 |
| auto-soft | 86.47 | 48.58 | 49.43 | **42.40** | 25.78 | **46.41** |

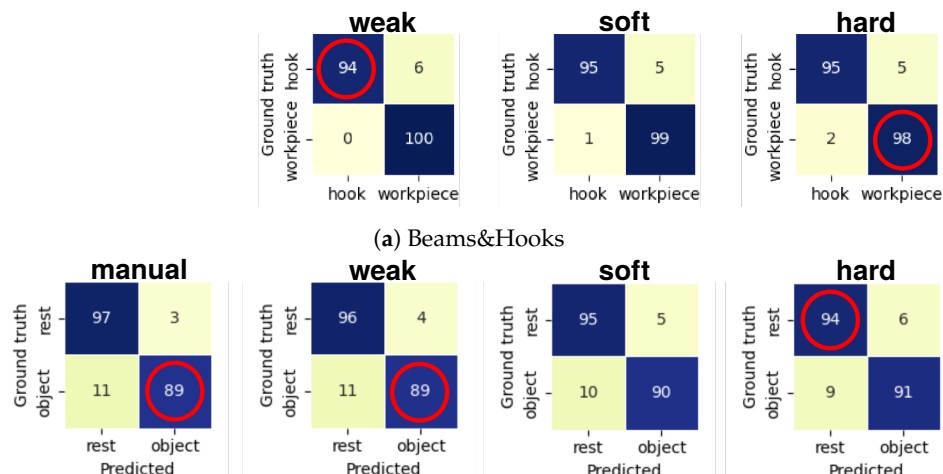

(**a**) Beams&Hooks

(**b**) Scan2CAD (validation set, 312 samples)—2-class (object vs. rest)

**Figure 15.** Comparison of 2-class segmentation performance across models as confusion matrices (normalized to show class-wise recall in diagonal) of predictions vs. ground truth. Worst scores per class across the models are circled in red.

Overall, these results show the viability of our automatic labels as cheap training data. Indeed, the PointNet++ segmentation performance on Beams&Hooks closely approaches the label performance from Table 3. While the task is more difficult for Scan2CAD, PointNet++ models trained on cheap labels are competitive compared with the expensive manual baseline model. Our results also highlight the difficulty of segmenting boundary areas, even when learning from manual labels. Comparing the performance achieved across the three automatic annotation schemes suggests that when training a point cloud segmentation model on approximate labels, weak or soft supervision (where the training loss is attenuated in ambiguous areas) is beneficial compared with conventional hard supervision (where the model is equally penalized for wrong predictions in ambiguous regions as in easy regions).

5.3.5. A Closer Look at Soft Predictions

So far, we have evaluated PointNet++ models only based on predicted classes compared with ground truth labels. However, in downstream tasks, it may be relevant to also consider the associated predicted confidence score. For instance, in the Beams&Hooks use case, the purpose of segmentation is to extract workpiece points, reconstruct a 3D model [57], and use this to instruct robot plans. If a prediction's confidence score is a reliable indicator of whether the point belongs to the object, ambiguous points could be excluded from the reconstruction by increasing the confidence threshold. As another example, in the context of scene understanding for robot navigation, it can be desirable to leave a safe margin around segmented obstacles [44]; this would require points around furniture in Scan2CAD to be assigned low but non-zero confidence scores.

We therefore take a closer look at the softmax logits predicted by the PointNet++ models trained on approximate labels. As visualized in Figure 16, we observe significant differences in predicted confidence scores for the object depending on the labeling scheme used during training (hard, weak, or soft). In predictions by the soft model, the object confidence score smoothly decreases along boundaries (e.g., the area where the hook and piece attach). In contrast, the hard model tends to confidently over-segment the workpiece, while the weak model outputs noisy/polarized predictions in boundary areas.

To quantify these observations, we threshold predictions at different levels, and for each object, we evaluate both the precision compared with manual ground truth (left of Figure 17) and the mean euclidean distance to the CAD model (right of Figure 17). Across both datasets, we find that shifting the confidence threshold unlocks a wider range of precisions and distances for the soft model than for the weak and hard models; that is, by increasing/reducing the

threshold used to binarize predictions, we can include/exclude an increasing number of points based on how likely they are to belong to the object. In contrast, the weak model's predictions have the lowest dynamic range and are the least expressive.

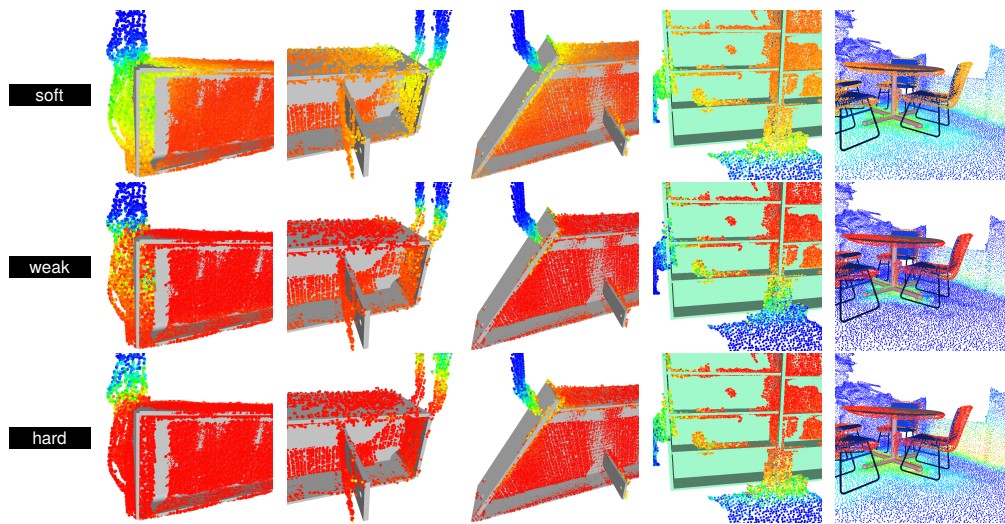

**Figure 16.** Visualization of CAD models and predicted logits for the "object" class by PointNet++ models. Samples are taken from the Beams&Hooks test set (top 3) and Scan2CAD validation set (bottom) and cropped for readability.

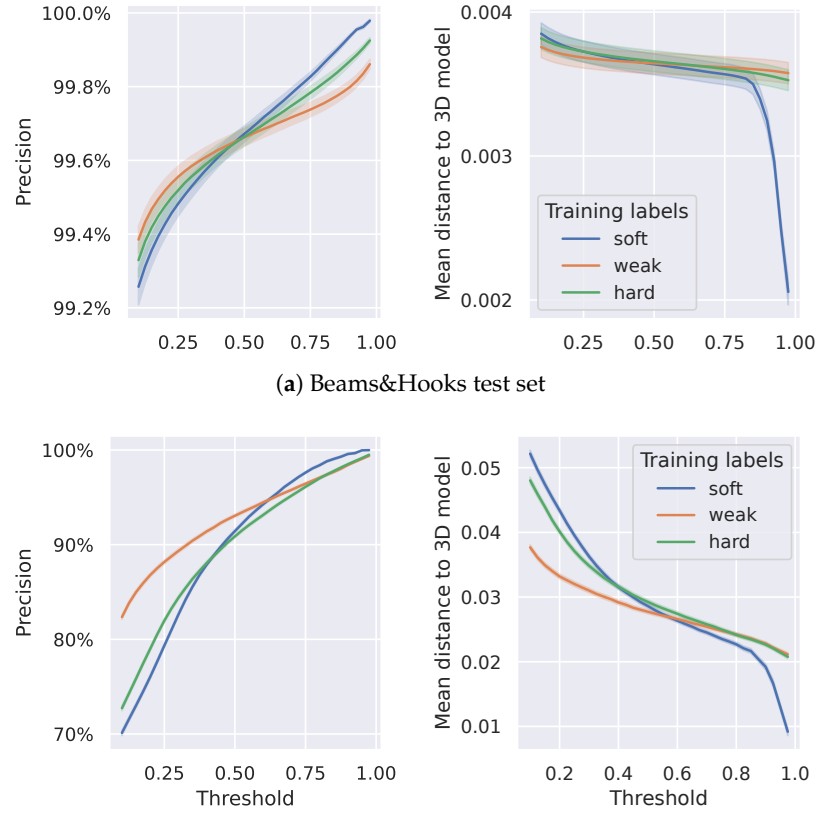

**Figure 17.** Precision (wrt. ground truth labels) and mean distance of object points to 3D model as a function of confidence threshold. Scores are computed per object; filled area represents the standard error across objects in the evaluation set.

## 6. Discussion

**Importance of CAD alignment**: Since our automatic labeling method exclusively relies on CAD-point cloud pairs to generate semantic labels, it is sensitive to both the quality of the 3D scans (noise and other artefacts make labels less accurate) as well as the quality of the CAD retrieval (how well does the CAD model match with the real object that was scanned) and alignment (how well is the CAD model fitted to the point cloud). While these steps are beyond the scope of this work, CAD retrieval and alignment can be particularly challenging when dealing with a large database of 3D models, in which there may not even be a perfect match. State-of-the-art approaches in 3D object retrieval aim not only to find good matches [58] and estimate their 9DOF pose [59], but also to estimate a deformation of the 3D object for a better fit in the scan [60]. A systematic analysis of the effect of the data acquisition set-up and CAD matching pipeline would provide further insights into the performance and applicability of our method.

**Better labels:** A notable limitation of the soft labeling method for multi-class datasets (e.g., Scan2CAD) is that it reduces the problem to an object-vs-background assignment without considering the labeling ambiguity between different objects that are next to each other (e.g., a keyboard sitting on a desk) or that are of similar type/geometry (e.g., a cabinet and dishwasher). Incorporating spatial and semantic relations between objects in soft labels is an important direction to explore. More generally, the modularity of the framework leaves room for incorporating other metrics or features than the three presented in this paper (region growth, distance, and SVM scores), which we hope can make it even more versatile and further improve label quality. For instance, we only consider geometric information, but it would be interesting to introduce color features (when available) in order to better inform object boundaries. Lastly, a natural extension of this work would be to label object instances based on CAD models; this would be valuable since instance segmentation annotations are even more difficult to come by and tedious to create manually.

**Learning from approximate labels:** We have employed a simple training scheme in our experiments (no data augmentation, consideration for class imbalance, etc.) and a single segmentation architecture (PointNet++), as the focus was on providing a relative comparison between labeling schemes rather than maximizing absolute performance. We would expect significant gains by employing the PointNet++ scaling and training strategies proposed in PointNext [47], for instance. Future work should extend this study to incorporate more recent architectures and other datasets to further analyze the performance and qualitative differences between labeling schemes.

In addition, for the weak labeling scheme, rather than simply ignoring unlabeled points in the loss during training, adding incomplete and inexact supervision losses [29] is a promising way to constrain learning for these points. Lastly, the segmentation model's performance is currently limited by the quality of the approximate labels. In the image domain, it has been shown that incorporating a small number of manually labeled samples as a fine-tuning step significantly improves performance while keeping annotation costs low [61]. This could help improve segmentation along boundaries, which are the most difficult to auto-label.

## 7. Conclusions

We have shown how CAD models can be leveraged at training-time for cheap 3D semantic segmentation labeling. Our pipeline extends Beams&Hooks and Scan2CAD with CAD-based semantic annotations at no cost and could easily be applied to other datasets for which CAD models can readily be obtained. We validate the label quality on real large-scale point clouds, showing that our method can cope with scanning imperfections (varying sampling density, noise, outliers, missing data) and even when only rough "generic" 3D models (rather than closely aligned industry-grade CADs) are available. Furthermore, we show that the labeling scores reliably capture labeling difficulty and can be learned by a deep segmentation network for more accurate and expressive predictions. Future research should investigate the relevance of training on soft semantic labels for downstream tasks.

**Author Contributions:** Conceptualization, G.H.-R., A.M. and S.B.J.; methodology, G.H.-R.; software, G.H.-R.; validation, G.H.-R.; formal analysis, G.H.-R.; investigation, G.H.-R.; resources, G.H.-R., A.M. and S.B.J.; data curation, G.H.-R. and S.B.J.; writing—original draft preparation, G.H.-R.; writing— review and editing, G.H.-R., A.M. and S.B.J.; visualization, G.H.-R.; supervision, A.M.; project administration, A.M.; funding acquisition, A.M. All authors have read and agreed to the published version of the manuscript.

**Funding:** This research was partly funded by Digital Lead (grant name: Point Cloud Intelligence).

**Data Availability Statement:** The Scan2CAD dataset was obtained from previously published work by Avetisyan et al. [13] and is publicly available at https://github.com/skanti/Scan2CAD (accessed on 13 July 2023). Sample data from the Beams&Hooks dataset presented in this study is available on request from the corresponding author. The full dataset is not publicly available due to company policies.

**Conflicts of Interest:** The authors declare no conflict of interest. The funders had no role in the design of the study; in the collection, analyses, or interpretation of data; in the writing of the manuscript; or in the decision to publish the results.

## Abbreviations

The following abbreviations are used in this manuscript:

| | |
|---|---|
| BIM | Building Information Modeling |
| CAD | Computer Aided Design |
| DoF | Degree of Freedom |
| IoU | Intersection over Union |
| LiDAR | Light Detection And Ranging |
| OA | Overall Accuracy |
| SVM | Support Vector Machine |

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
