# Peer review of "From CAD Models to Soft Point Cloud Labels: An Automatic Annotation Pipeline for Cheaply Supervised 3D Semantic Segmentation"

_remotesensing, doi:10.3390/rs15143578_

Round 1
Reviewer 1 Report
The paper “From CAD models to soft point cloud labels: An automatic annotation pipeline for cheaply supervised 3D semantic segmentation” (#remotesensing-2385833) aims to revise the annotation of 3D point cloud indoor scenes with soft labeling. The soft annotation downgrades with the binary semantic labels by starting with 3D CAD models fitted into the settings and disregarding the connection areas around joints from known labels to “other” or background. With the soft label, the mIoU score of PointNet++ has increased by about 2%** in the author’s dataset and 1~2%** in Scan2CAD dataset. Overall, I am deeply concerned by three significant issues: problematic research design and tests, outdated baseline DL, and limited contribution to the knowledge.
(** Note: the population of point/object targets has changed, So comparative values reported were meaningless.)
A rejection is recommended for the editor’s consideration because I do not see a possibility of correcting or defending all the issues in one or two rounds of revisions.
1. Failed research design and tests
The research design paper reads like a “(re-)painting the bullseye around the shot arrow” study. It may work in a confined set of indoor scenes for a while. Fundamentally, the soft annotation partially deletes truth information and thus undermines the semantic capability of the dataset. But publishing such a data engineering trick as a methodology in the target reputed journal is inappropriate.
The “arrow” here is PointNet++, a well-known deep CNN algorithm for processing 3D point cloud scenes. The arrow was shot (trained and tested on the datasets) but not hitting the gold central area on the target face (less satisfactory performance). The usual response is to develop and shoot another arrow, say, novel transformer ANNs, in the literature.
However, this paper tries to change the bullseye (mIoU and a few metrics) by redefining the target classes. The paper downgrades the “likely-wrong” points from the semantic targets “appliances, basket, sofa, etc.” to “rest/background.” As a result, the population of the targets changed. As reported, the mIoU metric showed a 1~2% change, which was inappropriate to support the novelty and soundness of the presented annotation.
Unless other consistent (on the same gold area on the targets) performance metrics are provided, the readers are not convinced that the annotation improved PointNet++.
2. Outdated baseline DL
PointNet++ is well-known but outdated as well. For example, its mIoU for S3DIS instance segmentation is 47%, according to the leader board at https://paperswithcode.com/sota/semantic-segmentation-on-s3dis. On the top of the list, many new algorithms, including transformer architectures, show much stronger competitiveness. Examples are Swin3D-L, PointNeXt-XL, and RepSurf-U, which improved PointNet’s mIoU from 47% to 79% (by +32%). Besides, most of the new DL algorithms are open-sourced as well.
The authors are recommended to test the new DL algorithms and PointNet++.
3. Limited contribution to the domain of knowledge of remote sensing
The annotation method is limited in contributing to the field of remote sensing. Instead of arguing a novel annotation, the authors may restructure the paper as an explanation of the limited capacity of CNN (PointNet++) in processing 3D points as the visualized object scores in Figure 5 and consider other journals focusing on deep learning theories, such as IJML, Neurocomputing, and Evolutionary Computation.
I recommend rejecting this manuscript for publication due to the above issues. The study requires significant redesign and rewriting to meet the journal's standards.
4. Minor issues
- “GT” is often known as the abbreviation of ground truth. The authors may consider using “baseline” or another name in Sect. 5.
- OA and mAcc are biased performance metrics. Please use unbiased ones such as mIoU and macro F1.
- Proofreading is required to eliminate some typos and grammar issues. E.g., there is a mismatch between the actor “our … method” and leading action “look at” in “Looking at the confusion matrix in Figure 10, our binary labeling method has …”
Excellent in general.
Proofreading is required to eliminate some typos and grammar issues. E.g., there is a mismatch between the actor “our … method” and leading action “look at” in “Looking at the confusion matrix in Figure 10, our binary labeling method has …”
Reviewer 2 Report
This paper describes a interesting comprehensive automated annotation scheme that leverages a raw 3D point cloud and a set of fitted CAD models as input. The scheme generates point-wise labels that can be utilized as cost-effective training data for point cloud segmentation. The results presented in the paper demonstrate a good accuracy of the automatic labels and a significant reduce annotation time, when compared to manual annotations.
The paper can be accepted with the following minor revisions:
1. Lines 47-49: Even if it is claimed that the 3D model retrieval & alignment step are beyond the scope of this work, a paragraph describing this process with more recent references would be very helpful for the reader.
2. Paragraph 2.1 Model-based segmentation: Related work in 3D point cloud segmentation should be enriched with more recent references.
3. Figures 2,3,4. The 3D point cloud acquisition method and the parameters of the 3D scanning procedure in the examples shown in these figures should be described, since this plays an important role to the accuracy of the produced models.
4. Furthermore please explain how the accuracy of the 3D scanning procedure affects the results of the proposed methodology
5. Table 2. Please explain what kind of unique CADs are included in these 2 datasets (149 and 3049 respectively)
6. Please add some more explanations for the confusion matrix in figure 10. An example could be very helpful
7. Figure 13: Use identical y axis labels in both cases in order to enhance comparison between them
8. The abbreviation SVM should be analyzed in its first appearance in the manuscript
Round 2
Reviewer 1 Report
I appreciate the responses and careful revisions in the latest version of the paper. On the one hand, the clarity of the core concepts has been considerably improved in the presentation. On the other hand, however, the three fatal issues, i.e., problematic research design, outdated DL, and limited contribution to remote sensing, are unresolved.
1. Failed research design and tests
Thanks to the improved clarity in the revised version, I agree with the authors that the population in the statistics remains the same. But, the major issue of partially deleting the truth information is not resolved.
Remote sensing readers care about the accurate signal stimuli in sensors and factual geometry with object-level semantics. Given the true stimuli and reference CAD models, auto-hard and auto-weak annotations have their sound and theoretical grounds for maintaining the object-level semantic capability of the dataset. The proposed auto-soft annotation failed me from the point-level orientation to scaled-up experiments.
For example, the 3D points in Figure 4 represented factual signals and accurate geometry of a soft and fluffy armrest of a true sofa – but the so-called “ground truth” CAD was another model. In the hard annotation, the points nearby the armrest are labeled as “[close to the CAD of] sofa,” or as “Rest [not the exact CAD in the dataset]” in the weak scheme. Both annotations keep the object-level semantics in the processes. The soft annotation adds the probabilities to the hard scheme, like “90% [close to the CAD of] sofa” and “10% [close to the CAD of] armchair,” after lengthy and black-box machine learning approximation. As a result, the scenes of real-world objects are transformed into fragmented and disconnected points.
The experimental results confirmed my perspective. The weak annotation showed the best object-level F-1 scores by mapping to the exact CADs in both datasets. The proposed soft annotation failed in the results. The reason, to me, is that the soft annotation deletes the innate continuity of real-world objects for fragmented probabilities. This may be an engineering trick for a local region of marginal points, but inappropriate for remote sensing of physical objects.
2. Outdated DL. The authors agreed that PointNet++ is outdated. Citing works in irrelevant research designs (tree simulator https://doi.org/10.3390/rs15092380; as a baseline to attack in https://doi.org/10.3390/rs15040982 ) cannot justify the use in this paper.
3. Limited contribution to remote sensing. Once again, I feel that the soft annotation is perhaps novel in terms of point-level geometry analysis, -- but in the fields of computer graphics or machine learning rather than Remote Sensing. I recommend the authors seriously consider submitting the right paper to the right journals, say, ACM ToG or IJCV, for the interest of the readers and the paper itself.
